# Future Motion Dynamic Modeling via Hybrid Supervision for Multi-Person Motion Prediction Uncertainty Reduction

Yan Zhuang
Fudan University
Shanghai, China
zhuangy23@m.fudan.edu.cn

Yanlu Cai
Fudan University
Shanghai, China
ylcai20@fudan.edu.cn

Weizhong Zhang*
Fudan University
Shanghai, China
weizhongzhang@fudan.edu.cn

Cheng Jin*
Fudan University
Shanghai, China
jc@fudan.edu.cn

## Abstract

Multi-person motion prediction remains a challenging problem due to the intricate motion dynamics and complex interpersonal interactions, where uncertainty escalates rapidly across the forecasting horizon. Existing approaches always overlook the motion dynamic modeling among the prediction frames to reduce the uncertainty, but leave it entirely up to the deep neural networks, which lacks a dynamic inductive bias, leading to suboptimal performance. This paper addresses this limitation by proposing an effective multi-person motion prediction method named Hybrid Supervision Transformer (HSFormer), which formulates the dynamic modeling within the prediction horizon as a novel hybrid supervision task. To be precise, our method performs a rolling predicting process equipped with a hybrid supervision mechanism, which enforces the model to be able to predict the pose in the next frames based on the (typically error-contained) earlier predictions. Addition to the standard supervision loss, two self and auxiliary supervision mechanisms, which minimize the distance of the predictions with error-contained inputs and the predictions with error-free inputs (ground truth) and guide the model to make accurate predictions based on the ground truth, are introduced to improve the robustness of our model to the input deviation in inference and stabilize the training process, respectively. The optimization techniques, such as stop-gradient, are extended to our model to improve the training efficiency. Furthermore, we develop a fine-grained spatio-temporal correlation capture module to assist the feature learning and reduce the uncertainties arising from the intricate and varying interactions among the individuals. Our approach achieves state-of-the-art results on multiple multi-person datasets in both short- and long-term prediction.

## CCS Concepts

• **Computing methodologies** → **Computer vision tasks**.

## Keywords

Human motion prediction, Rolling prediction, Hybrid supervision

---

*Weizhong Zhang and Cheng Jin are co-corresponding authors.

**ACM Reference Format:**
Yan Zhuang, Yanlu Cai, Weizhong Zhang, and Cheng Jin. 2024. Future Motion Dynamic Modeling via Hybrid Supervision for Multi-Person Motion Prediction Uncertainty Reduction. In *Proceedings of the 32nd ACM International Conference on Multimedia (MM '24), October 28-November 1, 2024, Melbourne, VIC, Australia.* ACM, New York, NY, USA, 10 pages. https://doi.org/10.1145/3664647.3681528

## 1 Introduction

Human motion prediction is a crucial task, which predicts future motion trends based on the previous observations, and has wide applications in the fields of 3D character animation [5, 7, 25, 44, 46], surveillance systems [6, 22, 52], and autonomous driving [8, 24, 50]. Thanks to the Transformer's powerful capability in modeling sequential relationships, recent approaches[33, 42, 45], employing Transformers for multi-person interaction modeling and implicitly temporal relationship modeling, achieve promising results.

However, due to the significant motion dynamic uncertainty within the forecasting horizon, i.e., the steps with motion to be predicted, multi-person motion prediction remains a challenging problem. These uncertainties mainly arise from the following two sources. One is the intricate and varying interactions among the individuals. Figure 1b (1) shows that the left wrist joint of the player in red obstructs several joints in the torso area of the player in blue, while the left wrist joint of the player in blue obstruct the hip joint joint of the player in red. The other is the perplexing evolution patterns in human motions. For example, the motion characteristics of different keypoints differ greatly. As shown in Figure 1b (2), the movement range of the player's knee joint is much smaller than that of the ankle joint. More importantly, the uncertainties can grow rapidly over the forecasting horizon. In this work, we argue that explicitly modeling the motion dynamic within the forecasting horizon should be an indispensable module to reduce the uncertainty and finally improve the motion prediction accuracy, which is always overlooked in the previous studies as they usually leave it entirely up to the decoder neural networks.

To address the above issues, we propose a novel Transformer based framework Hybrid Supervision Transformer (HSFormer). Our key idea is to explicitly model the motion dynamics in the forecasting horizon by constructing the interdependence among the adjacent frames. We first propose a rolling predicting process, which predicts the poses in the next frames based on the (typically error-contained) earlier predictions. Notice that training the model under this framework is highly nontrivial. For example, with vanilla optimization algorithms, the model would be susceptible to the accumulated prediction errors in the previous prediction. Therefore, addition to the standard supervision loss, which minimizes the distance between the ground truth and our rolling prediction

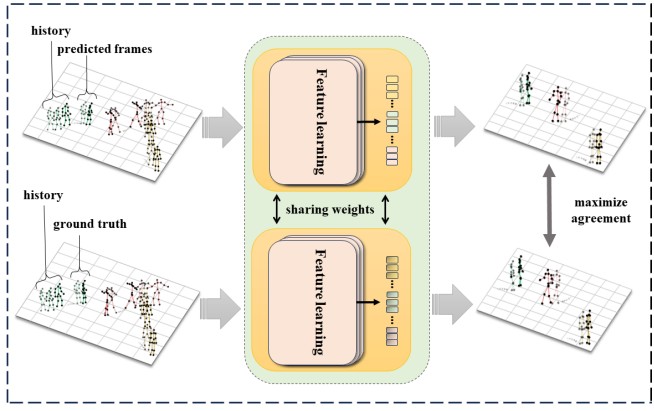

(a) The process of hybrid supervision.

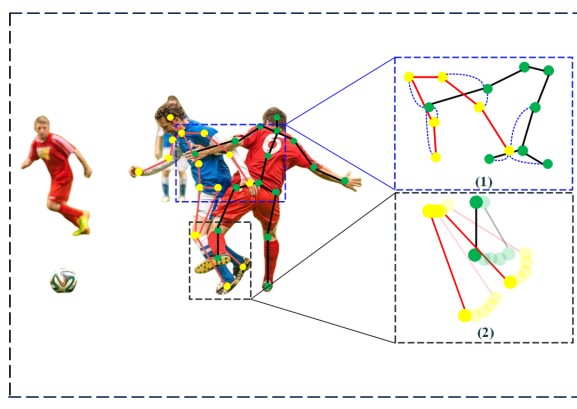

(b) Interactions of local joints among players.

Figure 1: Example of hybrid supervision and detailed interactions between individuals.

output, two additional supervision mechanisms, i.e., the self and auxiliary supervision mechanisms, are introduced to improve the robustness of our model to the input deviation in inference and stabilize the training process, respectively. To be precise, the self supervision mechanism minimizes the distance of the predictions with error-contained inputs and the predictions with error-free inputs (ground truth), while the auxiliary supervision mechanism guides the model to make accurate predictions based on the ground truth. These standard, self and auxiliary supervision mechanisms together form our novel hybrid supervised training framework. Additional optimization techniques, such as stop-gradient and parameter sharing, are extended to our HSFormer to improve the training efficiency.

Additionally, in order to reduce the uncertainties arising from the intricate and varying interactions, HSFormer utilizes a feature learning component, namely Spatio-Temporal Encoder (STE) block, for fine grained modeling. The STE block consists of the Spatial Joint Encoder (SJE) module and the Temporal Joint Encoder (TJE) module, which is responsible for capturing the relationships between body joints within each individual frame and the dynamic evolution of the joints, respectively.

We evaluate our method on 5 different datasets with varying scales and complexities. The results demonstrate the significant superiority of our HSFormer over the state-of-the-art methods. Notably, our method outperforms the current state-of-the-art approaches for both short-and long-term predictions, with 1%~23% accuracy improvement for the short-term (0s~1.0s) and 2%~8% accuracy improvement for the long-term prediction (1.0s~3.0s).

The main contributions of this work can be summarized as follows: 1) We design a rolling prediction scheme to effectively reduce the prediction uncertainty, which explicitly models the motion dynamic within the forecasting horizon by learning the interdependence between the motions in adjacent frames. 2) As training the model under our rolling prediction scheme is highly nontrivial, we develop a hybrid supervision mechanism to guarantee the training efficiency and the generalization capability of the learned

model. 3) We introduce the STE block, specifically designed to address the uncertainty arising from interaction details. STE enables a meticulous learning of the body joint relations in each frame and the global temporal correlation of each joint. 4) The experimental results on multiple multi-person motion datasets demonstrate that the proposed model significantly outperforms state-of-the-art methods.

## 2  Related work

### 2.1  Single-Person Pose Prediction

Single-person motion prediction [10, 11, 20, 22, 27, 37, 39, 41] involves the task of predicting the future motion of an individual based on historical motion. Compared to multi-person motion, single-person motion prediction has relatively simpler spatiotemporal dependencies and can be effectively captured using model architectures based on Recurrent Neural Networks (RNNs) [12–15, 17, 19, 26, 29, 32, 36, 49]. RNNs have been proven successful in sequence-to-sequence prediction tasks [21]. However, due to frame-by-frame prediction, issues of discontinuity and error accumulation often arise as the model independently generates predictions for each frame without considering the continuity with preceding and subsequent frames. To address the problem of error accumulation when using RNN-based models for long-term prediction, some research [4, 23] start exploring the use of fully connected or convolutional networks to better capture long-term dependencies and reduce error accumulation. In addition to RNNs, temporal convolutional networks have also shown promising results in modeling long-term motion [3, 4, 23, 28]. For instance, [23] construct a convolutional sequence-to-sequence model for human motion prediction. Unlike previous chain-based RNN models, the hierarchical structure of convolutional neural networks enables them to naturally model and learn spatial dependencies and long-term temporal dependencies. However, human motion is influenced by the surrounding environment, making it inherently uncertain, which becomes more evident in long-term prediction. Recent research begin to address this issue by jointly predicting human pose

and world coordinate trajectories [6, 43, 47, 48, 51]. For example, [6] leverage scene context to tackle the challenges of long-term prediction. Currently, single-person motion prediction has achieved promising results, while multi-person motion prediction is more complex due to factors such as crowd interaction. Our work extends to the simultaneous prediction of multi-person motion, including 3D pose and trajectories.

## 2.2 Multi-Person Pose Prediction

Multi-person motion prediction aims to predict the movement among multiple individuals in a scene [31, 35, 38]. This task requires considering interactions between individuals, making it more complex in a multi-person context compared to single-person motion prediction. The challenges arise because individual movements are influenced not only by their own dynamics but also by dynamic interactions with other individuals. To solve above issues, [1] introduce multi-person motion prediction, proposing a novel approach based on graph attention networks to model the dynamics of trajectories and poses, simulating interactions between individuals in both input space and output space. [42] introduces local-range encoders for individual motions and global-range encoders for social interactions. [16] introduce a cross-interaction attention mechanism that utilizes the historical information of two individuals, learning to predict cross-dependencies between two pose sequences. [33] introduce TBIFormer, proposing a novel Social Body Interaction Multi-Head Self-Attention (SBI-MSA) that learns dynamic body part interactions inter and intra individuals, capturing complex interaction dependencies. However, these methods predict all future frames directly from the input sequence, neglecting the dynamic changes within the predicted frames, which increases the uncertainty in the prediction process, especially in long-term predictions. In this paper, we propose a new multi-person motion prediction model framework based on Transformer to address the aforementioned issue.

## 3 Preliminaries

### 3.1 Problem Formulation

Given the historical motion trajectory of multiple individuals, our objective is to predict the future motion for these individuals. Formally, with $N$ individuals in the scene, the history motion trajectory of the $n$-th individual with $T + 1$ time steps can be presented as $X_{1:T+1}^n = [x_1^n, x_2^n, \ldots, x_{T+1}^n]$ with each $x_i^n \in \mathbb{R}^{J \times 3}$ being the coordinates of $J$ skeleton joints. Following the related work [42], we transform $X_{1:T+1}^n$ into $Y_{1:T}^n = [y_1^n, \ldots, y_T^n]$ by taking the difference between the adjacent entries to expose the motion trends to the model, i.e., $y_i^n = x_{i+1}^n - x_i^n$, $i = 1, \ldots, T$. Our goal is to predict the 3D pose sequence of the next $F$ time steps, i.e., $\hat{Y}_{T+1:T+F}^n = [\hat{y}_{T+1}^n, \ldots, \hat{y}_{T+F}^n]$, and transform it back to $\hat{X}_{T+2:T+F+1}^n$ with $n = 1, \ldots, N$.

### 3.2 Pipeline

Given a 3D pose sequence $Y \in \mathbb{R}^{N \times J \times T \times 3}$ with N persons, J joints, and T frame as input, the vanilla pipeline apply a Discrete Cosine Transformation (DCT) [2] to encode motion into the frequency domain, creating a more compact representation. This representation is then projected into a high-dimensional feature space to

obtain the embedded feature $\mathcal{F}_{Embed} \in \mathbb{R}^{N \times J \times T \times C}$, where $C$ is the feature dimension. Subsequently, $\mathcal{F}_{Embed}$ is fed into an encoder to learn sequence features followed by a decoder for predicting future sequences. Finally, the pipeline utilizes a fully connected (FC) layer and an Inverse Discrete Cosine Transformation (IDCT) to obtain the future motion $\hat{Y}_{T+1:T+F}$ for each individual. In this work, we reconstruct both the encoder and decoder block to reduce the prediction uncertainty, which are the main contributions of this paper. To be precise, we propose

- **Fine-grained Correlation Learning (see Section 4):** We propose a Spatio-Temporal Encoder to assist the fine-grained feature learning to finally reduce the prediction uncertainty arising from the intricate and varying interactions among the individuals. It can learn the joint relations in each frame and the global temporal correlation for each joint in different frames.
- **Dynamic Modeling via Hybrid Supervision (referring to Section 5):** We propose a rolling prediction scheme to explicitly learn the interdependence among the motions in adjacent frames. We further design a novel hybrid supervision mechanism to guarantee both the training efficiency and the robustness of the learned model to avoid prediction error accumulation.

## 4 Fine-grained Correlation Learning

For the convenience of presenting our method HSFormer, we first introduce our fine-grained correlation learning block STE as it is integrated into both the encoding and decoding components. STE block is designed to assist the feature learning to reduce the uncertainties arising from the intricate and varying interactions among the individuals. As shown in Figure 2, it consists of a SJE module and a TJE module, which learn the joint relations in each frame and the global temporal correlation for each joint in different frames, respectively.

**Spatial Correlation Learning**. SJE aims to learn the spatial relationships across body joints for all individuals via attention mechanism. Suppose the SJE module consists of $L_1$ layers, we embed $\mathcal{F}_{Embed}$ with the learnable position embedding $E_{spos} \in \mathbb{R}^{(N \times J) \times C}$ to obtain the embedded feature $\mathcal{F}_{SJE}^0 \in \mathbb{R}^{T \times (N \times J) \times C}$ before the first layer of SJE module. SJE module takes $N \times J$ tokens of size C from a certain frame as inputs and models relationship across all joints. After this, tokens will have knowledge about the joints of other individuals in the same frame. The output of the final layer can be represented as $\mathcal{F}_{SJE}^{L_1}$.

**Temporal Correlation Learning**. Due to the diversity of joint movements, it is crucial to learn the independent characteristics of each joint. The TJE module treats the motion trajectory of each joint as an independent unit, capable of identifying the unique motion characteristics of each joint. We first reshape the output feature $\mathcal{F}_{SJE}^{L_1}$ from SJE into $\mathbb{R}^{(N \times J) \times T \times C}$ and then combine it with the learnable temporal positional encoding $E_{tpos} \in \mathbb{R}^{(N \times J) \times T \times C}$ to obtain the feature $\mathcal{F}_{TJE}^0$. It is then fed into TJE module to learn the contextual dependencies for each joint in parallel. After processed by our TJE module, $\mathcal{F}_{TJE}^0$ is transformed into the final output $\mathcal{F}_{TJE}^{L_2}$, where $L_2$ is the number of TJE layers.

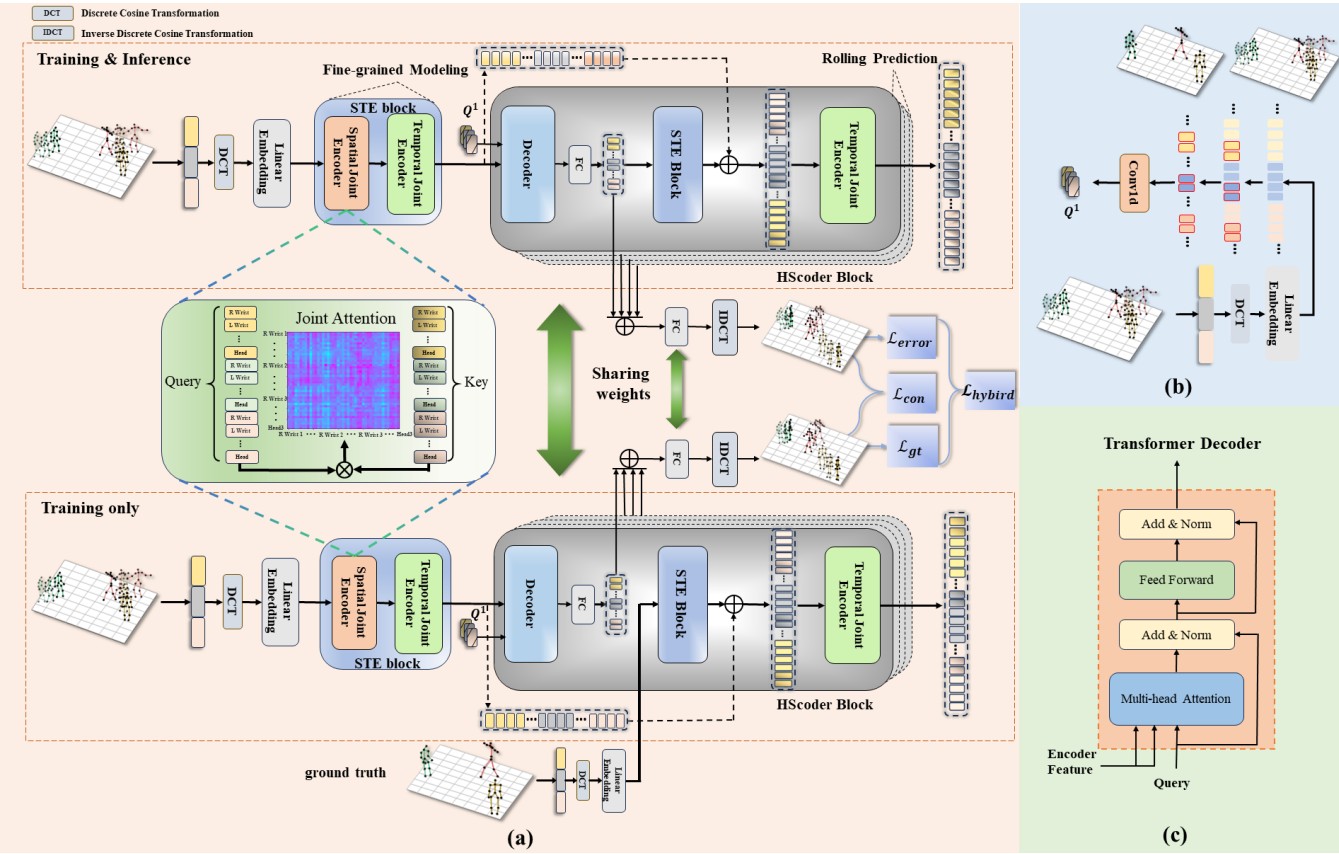

**Figure 2: (a) Overview of the proposed HSFormer framework. We reconstruct both the encoder and decoder in the conventional pipeline to reduce the prediction uncertainty. Specifically, we propose the STE block to capture the fine-grained spatio-temporal correlations among the joints to assist the feature learning. HScoder performs a rolling prediction process to learn the interdependence among the frames with the forecasting horizon. We further develop a hybrid supervision mechanism to guarantee the training efficiency and robustness of our model to the accumulated prediction error. (b) The process of constructing the body query token $Q_1$. (c) The standard Transformer Decoder used in HSFormer.**

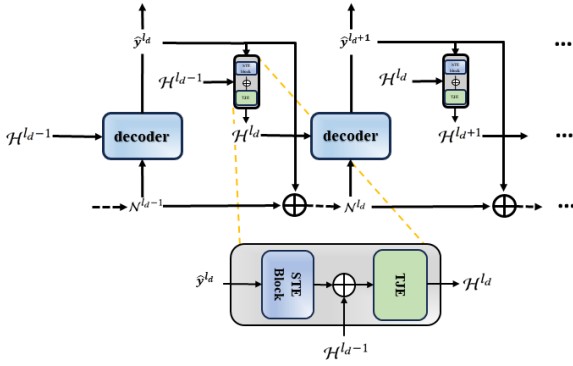

**Figure 3: Our Hybrid-Supervised coder (HScoder) block.**

## 5 Dynamic Modeling via Hybrid Supervision

### 5.1 Rolling Prediction Scheme

We reconstruct the decoder, referred to as the HScoder block, which is designed to guide the model to learn the dynamics of the predicted frames through a rolling prediction process to reduce uncertainty. As depicted in Figure 3, this process divides the forecasting horizon into multiple intervals and predicts the motions sequentially. To be precise, let $f$ be the length of the divided predicton interval, our rolling process would be comprised of $d = F/f$ prediction rounds. In the $l_d$-th round with $l_d \in \{1, \ldots, d\}$, our HScoder first constructs a body query token $Q^{l_d} \in \mathbb{R}^{N \times J \times C}$ based on the neighbor motion dynamics of the recent $\ell$ frames, denoted as $\mathcal{N}^{l_d} \in \mathbb{R}^{N \times \ell \times J \times C}$, that is,

$$Q^{l_d} = Conv1d(\mathcal{N}^{l_d}), \qquad (1)$$

where $Conv1d(\cdot)$ is a 1D Convolution layer with kernel size of $\ell$. Then HScoder leverages a Transformer Decoder to predict the future pose features of $f$ frames, denoted as $\hat{y}^{l_d} \in \mathbb{R}^{N \times f \times J \times C}$,

based on $Q^{l_d}$ and the guidance of the history context $\mathcal{H}^{l_d}$, that is,

$$\hat{y}^{l_d} = \text{FC}(Decoder(Q^{l_d}, \mathcal{H}^{l_d})), \qquad (2)$$

where $\text{FC}(\cdot)$ is linear layer to convert the single token into $f$ frames.

Our rolling prediction scheme takes $\mathcal{F}_{Embed}$ and $\mathcal{F}_{TJE}^{L_2}$ to obtain the initial neighbor motion dynamics $\mathcal{N}^1$ and history context $\mathcal{H}^1$, respectively. And then leverage the previous prediction $\hat{y}^{l_d}$ together with previous $\mathcal{N}^{l_d}$ and $\mathcal{H}^{l_d}$ to build the new ones, i.e., $\mathcal{N}^{l_d+1}$ and $\mathcal{H}^{l_d+1}$. Specifically,

$$\mathcal{N}^{l_d} = \begin{cases} Slice(\mathcal{F}_{Embed}), & l_d = 1 \\ Slice(Concat(\mathcal{N}^{l_d-1}, \hat{y}^{l_d-1})), & l_d > 1 \end{cases} \qquad (3)$$

where $Slice(\cdot)$ is to get the last $\ell$-frame feature. As for the history context $\mathcal{H}^{l_d}$, we re-encode the prediction $\hat{y}^{l_d}$ into the feature space of the history context by employing a STE module, that is,

$$\hat{z}^{l_d} = \text{STE}\left(\hat{y}^{l_d}\right). \qquad (4)$$

A TJE module is adopted after concatenation to extract the temporal relationship between previous history context and the new prediction. Thus, it can be formulated as

$$\mathcal{H}^{l_d} = \begin{cases} \mathcal{F}_{TJE}^{L_2}, & l_d = 1 \\ \text{TJE}(Concat(\mathcal{H}^{l_d-1}, \hat{z}^{l_d-1})), & l_d > 1 \end{cases} \qquad (5)$$

where $\mathcal{H}^{l_d} \in \mathbb{R}^{N \times (T+(l_d-1)*f) \times J \times C}$.

After $d$ rounds rolling prediction, all the prediction feature $\hat{y}^{l_d} \in \mathbb{R}^{N \times f \times J \times C}$ will be concatenated and projected to obtained the predicted pose sequence $\hat{Y}$, that is,

$$\hat{Y}_{T+1:T+F} = \text{IDCT}\left(\text{FC}\left(Concat\left(\hat{y}^1, \hat{y}^2, \ldots, \hat{y}^{l_d}\right)\right)\right), \qquad (6)$$

where the linear layer $\text{FC}(\cdot)$ is adopted to project the feature into the pose sequence, and $\text{IDCT}(\cdot)$ stands for Inverse Discrete Cosine Transformation. Finally, $\hat{Y}_{T+1:T+F}$ is transformed back to $\hat{X}_{T+2:T+F+1}$.

## 5.2 Hybrid Supervision Mechanism

Note that it is indeed not trivial to train such a network under rolling prediction framework, since the errors contained in previous predictions can accumulate over time, thereby adversely affecting subsequent predictions and potentially leading to the failure of training. This issue is particularly severe in the early training stage.

To solve the above issue, we propose a gt-augment branch together with a hybrid supervision mechanism, which provides a more distinct optimization path unaffected by the accumulation of errors, thereby enhancing the training stability. The details are presented as follows.

The gt-augmented branch is designed for training process only and will be removed in inference. It shares the same architecture and weights with the main branch to make the model in two branches be consistent, while it takes ground truth rather than previous predictions as inputs to predict the next frames. Specifically, it replaces the prediction feature $\hat{y}^{l_d}$ in Eqn. (3) and (4) with the feature $\hat{y}_{gt}^{l_d}$ embedded from the ground truth with $l_d \in \{1, \ldots, d\}$. The embedding process can be formulated as

$$[\hat{y}_{gt}^1, \ldots, \hat{y}_{gt}^{l_d}, \ldots, \hat{y}_{gt}^d] = \text{FC}\left(\text{DCT}\left(Y_{T+1:T+1+F}\right)\right), \qquad (7)$$

For simplicity, we let $X$, $\hat{X}$ and $\hat{X}_{gt}$ be the ground truth, the output of the main branch and the gt-augmented branch, respectively.

By denoting MSE-loss to be

$$\mathcal{L}_{\text{MSE}}(X, \hat{X}) = \frac{1}{N \times F \times J} \sum_{n=1}^{N} \sum_{i=T+2}^{T+F+1} \sum_{j=1}^{J} \left\| X_{i,j}^n - \hat{X}_{i,j}^n \right\|_2^2,$$

we train our two-branched model with the following hybrid supervision mechanism:

- **Standard Supervision.** It is adopted to enforce the model to be able to make accurate predictions even with typically error-contained previous predictions as inputs and the loss takes the form of $\mathcal{L}_{error} = \mathcal{L}_{\text{MSE}}(X, \hat{X})$.
- **Auxiliary Supervision.** To avoid the potential failure of training caused by error accumulation, especially in the early training stage, we adopt the following auxiliary supervision loss to guide the model to make accurate predictions taking the ground truth poses in the previous frames as inputs. That is $\mathcal{L}_{gt} = \mathcal{L}_{\text{MSE}}(X, \hat{X}_{gt})$.
- **Self Supervision.** A self-supervised contrastive loss $\mathcal{L}_{con}$ is adopted to minimize the distance of the prediction with error-contained inputs and the prediction with error-free inputs, enhancing the robustness of the model to the prediction errors in the previous frames, that is,

$$\mathcal{L}_{con} = \mathcal{L}_{\text{MSE}}(\hat{X}, sg(\hat{X}_{gt})),$$

where $sg(\cdot)$ stands for stop-gradient operation[9].

Thus, the overall loss becomes:

$$\mathcal{L}_{hybrid} = \lambda \mathcal{L}_{error} + (1 - \lambda)\mathcal{L}_{gt} + \gamma \mathcal{L}_{con}, \qquad (8)$$

where the parameters $\lambda \in (0, 1]$ and $\gamma > 0$ are adopted to tune the weights of the above three losses during training. We observe that our model is insensitive to $\gamma$ and we can achieve good performance with a common value $\gamma = 1$ in all our experiments. For $\lambda$, we gradually increase it from 0 to 1 during the training process for two reasons: 1) In the early training stage, a small $\lambda$ is preferred as the inputs of our first branch, i.e., the previous predictions, always contain pronounced errors, making the standard supervision mechanism ill posed; 2) Unlike our second branch does in training, we cannot make predictions by taking the ground truth as inputs in inference. Therefore, we require $\lambda$ to be close to 1 in the end of training to gradually discard our second branch to guarantee the consistency between the training and inference process.

## 6 Experiments

### 6.1 Implementation Details

Our experiments use PyTorch framework on two Nvidia GeForce RTX 3090 GPUs. We train with the batch size of 32 for 150 epochs. The learning rate is set to 0.0003. The number of stacked encoder layers, $L_1$ and $L_2$, are both set to 2, while the transformer decoder is stacked with 3 layers. The kernel size of the 1D Convolution layer $\ell$ is set to 10. For predictions within 1 second, $d$ is set to 5; for predictions within 1 to 3 seconds, $d$ is set to 9. $\lambda_1 = epoch_i/150$, where "$epoch_i$" represents the current epoch number.

**Table 1: The short-term prediction results of JPE, APE and FDE on the datasets CMU-Mocap, MuPoTS-3D, Mix1, and Mix2. We compare our method with the previous SOTA methods in 0.2 ~ 1.0 second. Best results are shown in boldface.**

| | | CMU-Mocap (3 persons) | | | | MuPoTS-3D (2-3 persons) | | | | Mix1 (6 persons) | | | | Mix2 (10 persons) | | | |
|---|---|---|---|---|---|---|---|---|---|---|---|---|---|---|---|---|---|
| | Time(s) | 0.2 | 0.6 | 1.0 | Average | 0.2 | 0.6 | 1.0 | Average | 0.2 | 0.6 | 1.0 | Average | 0.2 | 0.6 | 1.0 | Average |
| **JPE** | MRT [42] | 36 | 115 | 192 | 114 | 78 | 225 | 349 | 217 | 37 | 122 | 212 | 124 | 38 | 126 | 214 | 126 |
| | TBIFormer [33] | 30 | 109 | 182 | 107 | 66 | 200 | 319 | 195 | 34 | 121 | 209 | 121 | 34 | 118 | 198 | 117 |
| | SocialTGCN [34] | 28 | **96** | 163 | 96 | 86 | 214 | 324 | 208 | 46 | 126 | 210 | 127 | 68 | 130 | 199 | 132 |
| | JRFormer [45] | 38 | 118 | 178 | 111 | 124 | 276 | 383 | 261 | 37 | 126 | 222 | 128 | 28 | **104** | 185 | 106 |
| | Ours | **25** | 98 | **161** | **95** | **60** | **165** | **255** | **160** | **26** | 104 | **189** | 106 | **27** | 104 | **182** | **104** |
| **APE** | MRT [42] | 36 | 108 | 159 | 101 | 71 | 166 | 217 | 151 | 36 | 109 | 166 | 104 | 38 | 115 | 178 | 110 |
| | TBIFormer [33] | 27 | 84 | 118 | 76 | 60 | **132** | 170 | 121 | 28 | 81 | 113 | 74 | 30 | 89 | 124 | 81 |
| | SocialTGCN [34] | 26 | 79 | 115 | 73 | 84 | 166 | 242 | 164 | 43 | 103 | 143 | 96 | 63 | 108 | 147 | 106 |
| | JRFormer [45] | 33 | 96 | 131 | 87 | 93 | 173 | 212 | 159 | 30 | 79 | 112 | 74 | 28 | **82** | 124 | 78 |
| | Ours | **24** | **78** | **114** | **72** | **58** | 133 | **169** | 120 | **23** | **75** | **107** | **68** | 27 | 82 | **117** | **75** |
| **FDE** | MRT [42] | 27 | 88 | 157 | 91 | 59 | 187 | 309 | 185 | 29 | 100 | 189 | 106 | 29 | 98 | 185 | 104 |
| | TBIFormer [33] | 18 | 72 | 133 | 74 | 49 | 163 | 277 | 163 | 23 | 89 | 168 | 93 | 21 | 81 | 151 | 84 |
| | SocialTGCN [34] | 17 | 64 | **118** | 66 | 71 | 179 | 289 | 180 | 35 | 76 | **137** | 83 | 34 | 83 | 151 | 89 |
| | JRFormer [45] | 21 | 74 | 123 | 73 | 98 | 237 | 348 | 228 | 25 | 100 | 193 | 106 | 20 | 70 | **134** | 75 |
| | Ours | **14** | **61** | 120 | **65** | **39** | **113** | **195** | 116 | **16** | **74** | 149 | **80** | **16** | **69** | 136 | **74** |

**Table 2: The long-term prediction results of JPE, APE and FDE on the datasets CMU-Mocap, MuPoTS-3D, Mix1, and Mix2. We compare our method with the previous SOTA methods in 1.0 ~ 3.0 seconds. Best results are shown in boldface.**

| | | CMU-Mocap (3 persons) | | | | MuPoTS-3D (2-3 persons) | | | | Mix1 (6 persons) | | | | Mix2 (10 persons) | | | |
|---|---|---|---|---|---|---|---|---|---|---|---|---|---|---|---|---|---|
| | Time(s) | 1.0 | 2.0 | 3.0 | Average | 1.0 | 2.0 | 3.0 | Average | 1.0 | 2.0 | 3.0 | Average | 1.0 | 2.0 | 3.0 | Average |
| **JPE** | MRT [42] | 148 | 256 | 352 | 252 | 194 | 332 | 436 | 321 | 124 | 254 | 398 | 259 | 139 | 294 | 454 | 296 |
| | TBIFormer [33] | 118 | 225 | 329 | 224 | 189 | 321 | 432 | 314 | 117 | 242 | 374 | 244 | 116 | 232 | 346 | 231 |
| | SocialTGCN [34] | 102 | 205 | 310 | 206 | 229 | 374 | 523 | 375 | 109 | **232** | 376 | 239 | 112 | 226 | 341 | 226 |
| | JRFormer [45] | 122 | 218 | 305 | 215 | 196 | 334 | 430 | 320 | 110 | 248 | 380 | 246 | 106 | 220 | 336 | 221 |
| | Ours | **98** | **198** | **299** | **198** | **175** | **305** | **415** | **298** | **106** | 232 | **372** | **237** | **104** | **215** | **330** | **216** |

**Table 3: Performance on MI-Motion dataset for 5 different scenes. Best results are shown in boldface.**

| | | Park | | | Street | | | Indoor | | | Special Locations | | | Complex Crowd | | |
|---|---|---|---|---|---|---|---|---|---|---|---|---|---|---|---|---|
| | Time(s) | 0.32 | 0.56 | 1.0 | 0.32 | 0.56 | 1.0 | 0.32 | 0.56 | 1.0 | 0.32 | 0.56 | 1.0 | 0.32 | 0.56 | 1.0 |
| **JPE** | MRT [42] | 76 | 107 | 149 | 74 | 113 | 151 | 80 | 119 | 147 | 159 | 225 | 289 | 88 | 140 | 220 |
| | TBIFormer [33] | 64 | 96 | 141 | 60 | 96 | 131 | 69 | 108 | 154 | 158 | 236 | 312 | 63 | 104 | 158 |
| | SocialTGCN [34] | 60 | 95 | 154 | 54 | 81 | 124 | 67 | 108 | 160 | 165 | 246 | 322 | 70 | 113 | 177 |
| | JRFormer [45] | 47 | 81 | 134 | 57 | 92 | 102 | 75 | 95 | **120** | 191 | 278 | 331 | 56 | 98 | 152 |
| | Ours | **43** | **72** | **121** | **38** | **65** | **92** | **50** | **85** | 125 | **151** | **212** | **277** | **54** | **92** | **151** |

## 6.2 Experimental Setting

**Dataset.** Following the state-of-the-art method TBIFormer[33], we conduct our experiments on five widely-used datasets with varying scales and complexities. CMU-Mocap dataset consists of a training set with 6,000 sequences and a test set with 800 sequences. MuPoTS-3D (2 to 3 persons) [30] is a dataset of 3D human poses collected by a multi-view unmarked motion capture system. Mix1 (6 persons) and Mix2 (10 persons) are composed by blending data from CMU-Mocap, 3DPW [40], and MuPoTs-3D [30] datasets. On the above datasets, we follow TBIFormer [33] for fair comparison, i.e., the model takes 50 frames (2.0s) as inputs to predict 25 frames (1.0s) for short-term prediction, while the model takes 15 frames as inputs (1.0s) to predict 45 frames (3.0s) for long-term prediction.

The MI-Motion dataset is a large-scale dataset, which includes 167k frames of multi-person skeleton poses and covers five different everyday activity scenarios: indoor, park, street, special locations, and crowded scenes. On MI-Motion dataset, the model takes 25 frames (1.0s) as inputs to predict 25 frames (1.0s), which is consistant with SocialTGCN [34].

**Baseline.** We choose the most recent state-of-the-art multi-person prediction methods as baselines, i.e., MRT [42], TBIFormer [33], SocialTGCN [34] and JRFormer [45].

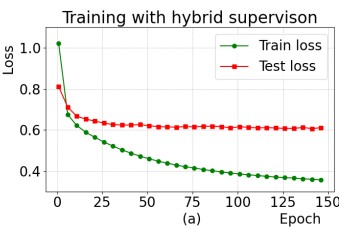 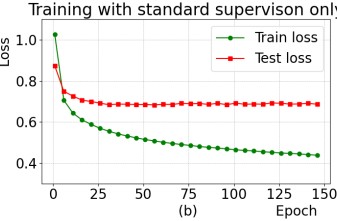 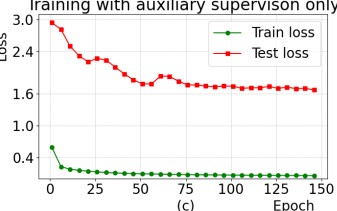

**Figure 4: Comparison of training loss and test loss on the CMU-Mocap dataset under three different conditions: hybrid supervision, i.e., $\mathcal{L}_{hybrid}$, standard supervision only, i.e., $\mathcal{L}_{error}$ and auxiliary supervision only, i.e., $\mathcal{L}_{gt}$.**

**Table 4: Ablation experiments with different rolling numbers show the results of our model on the CMU-Mocap in JPE metric.**

|  | JPE | | | APE | | | FDE | | |
|---|---|---|---|---|---|---|---|---|---|
| rounds | 1.0s | 2.0s | 3.0s | 1.0s | 2.0s | 3.0s | 1.0s | 2.0s | 3.0s |
| d=1 | 103 | 205 | 316 | 83 | 129 | 158 | 69 | 158 | 259 |
| d=3 | 102 | 207 | 310 | 80 | 127 | 149 | 66 | 158 | 255 |
| d=5 | 101 | 205 | 308 | **79** | 127 | 151 | 65 | 154 | 252 |
| d=9 | **98** | **198** | **299** | **79** | **123** | **145** | **62** | **149** | **243** |
| d=15 | 102 | 206 | 308 | 83 | 127 | 150 | 66 | 155 | 251 |

**Table 5: Ablation studies on different components of CONS-Former. Our full method and its variants are evaluated on the CMU-Mocap in JPE metric.**

|  | JPE | | | APE | | | FDE | | |
|---|---|---|---|---|---|---|---|---|---|
| Method | 1.0s | 2.0s | 3.0s | 1.0s | 2.0s | 3.0s | 1.0s | 2.0s | 3.0s |
| w/o Rolling Design | 103 | 205 | 316 | 83 | 129 | 158 | 69 | 158 | 259 |
| w/o Hybrid Supervision | 104 | 208 | 314 | 85 | 129 | 156 | 70 | 159 | 257 |
| w/o SJE | 103 | 206 | 306 | 84 | 130 | 154 | 69 | 160 | 255 |
| w/o TJE | 104 | 209 | 308 | 82 | 129 | 153 | 67 | 158 | 257 |
| Full | **98** | **199** | **298** | **79** | **123** | **145** | **62** | **149** | **243** |

**Metrics.** We evaluate our results with following three metrics, which are consistent with those in TBIFormer [33]. The detailed definitions of these metrics are given below:

**1) JPE.** We use the Mean Per Joint Position Error (MPJPE) [18] metric to measure the poses of all the individuals, including body trajectory:

$$\text{JPE}(X, \hat{X}) = \frac{1}{N \times F \times J} \sum_{n=1}^{N} \sum_{i=T+2}^{T+F+1} \sum_{j=1}^{J} \left\| X_{i,j}^n - \hat{X}_{i,j}^n \right\|_2,$$

where $X_{i,j}^n$ and $\hat{X}_{i,j}^n$ are the ground truth and the prediction of the $j$-th joint for the $n$-th individual in the $i$-th frame.

**2) APE.** We remove global movement and use Aligned mean per joint Position Error (APE) to measure pure pose position error:

$$\text{APE}(X, \hat{X}) = \frac{1}{N \times F \times J} \sum_{n=1}^{N} \sum_{i=T+2}^{T+F+1} \sum_{j=1}^{J} \left\| \left( X_{i,j}^n - X_r^n \right) - \left( \hat{X}_{i,j}^n - \hat{X}_{i,r}^n \right) \right\|_2,$$

where $X_{i,r}^n$ and $\hat{X}_{i,r}^n$ are the ground truth and prediction of the root joint for the $n$-th individual in the $i$-th frame.

**3) FDE.** In order to evaluate the global motion of all individuals, we introduce the Final Displacement Error (FDE), that is

$$\text{FDE}\left(X_{T+F+1,r}, \hat{X}_{T+F+1,r}\right) = \frac{1}{N} \sum_{n=1}^{N} \left\| X_{T+F+1,r}^n - \hat{X}_{T+F+1,r}^n \right\|_2,$$

where $X_{T+F+1,r}^n$ and $\hat{X}_{T+F+1,r}^n$ represent the ground truth and prediction of the root joint at the last frame, i.e., the $(T + F + 1)$-th frame, for the $n$-th individual.

## 6.3 Evaluation Results

**Quantitative Results.** Table 1 presents the results of JPE, APE and FDE on the datasets CMU-Mocap, MuPoTS-3D, Mix1 and Mix2 (cross-dataset validation) in short term (0.2s~ 1.0s). Our model

provides rolling guidance for the prediction sequence of multiple frames and establishes long-term temporal dependencies. Compared to the previous best method, our model demonstrates an improvement in the JPE metric ranging from 1% to 23% in the four datasets. Furthermore, Table 2 validates the effectiveness of our model over longer time intervals (1.0s ∼ 3.0s). Compared to the previous best method, our model exhibits an improvement of 2% ∼ 8%. Additionally, as shown in Table 3, we evaluate the performance of our model on the large-scale dataset MI-Motion. Our model outperforms previous model, with improvement of 3% ∼ 22%, in terms of the JPE metric across five different scenes.

**Qualitative Results.** Figure 5 shows our visualization results compared to the baselines and ground truth. TBIFormer [33] demonstrates the trend of converging to a static pose in long-term predictions, with less pronounced motion amplitudes. JRFormer [45] does not perform well in learning the climbing pattern, as indicated by the red and black boxes in the figure. In contrast, our model generate dynamic and accurate prediction results. Compared to other methods, our approach is closer to the ground truth.

## 6.4 Ablation Studies

We further conduct extensive ablation studies on CMU-Mocap to investigate the contribution of technical components in CONSFormer, with results given in Table 4 and Table 5.

**Effectiveness of Rolling Design.** The purpose of Rolling Design is to provide rolling guidance for the prediction sequence of multiple frames and establish long-term temporal dependencies. As illustrated in the first row of Table 5, if the Rolling Design is removed, our model exhibits more pronounced errors in long-term predictions. Additionally, in Table 4, we conduct experiments on the number of rolling predictions and find that predicting 5 frames per roll (d=9) yields the best results in long-term predictions.

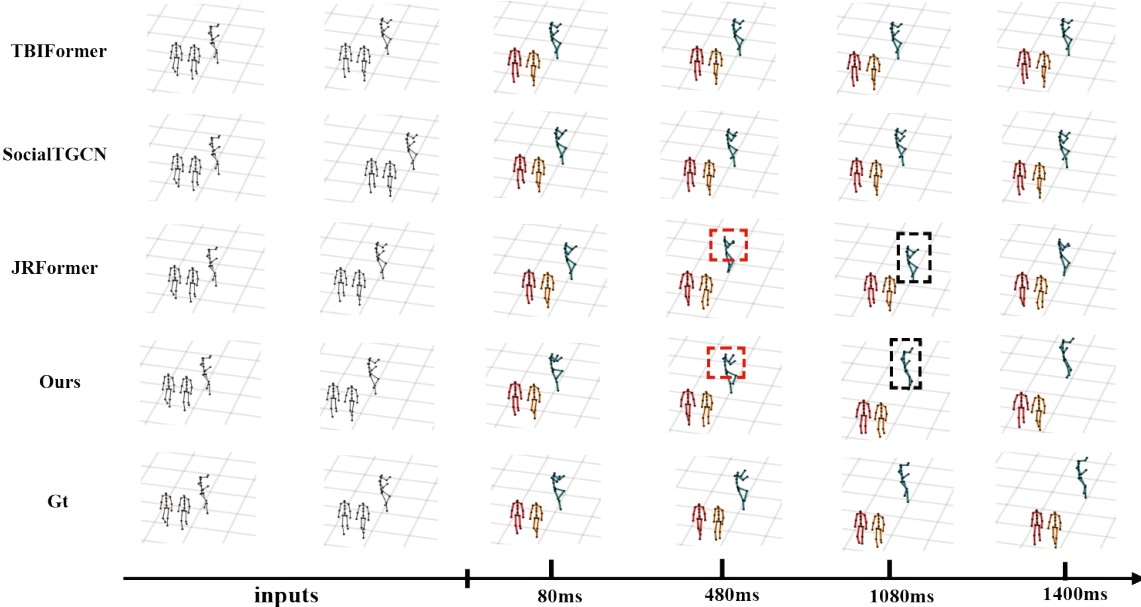

**Figure 5: Visualization comparison with the baselines and the ground truth on the sample of the MI-Motion dataset. The left two columns are inputs, and the right four columns are predictions. Our method successfully captures the alternating motion pattern of the legs when a person climbs a ladder. In contrast, other methods only capture the overall upward movement, keeping the legs in their predictions nearly stationary.**

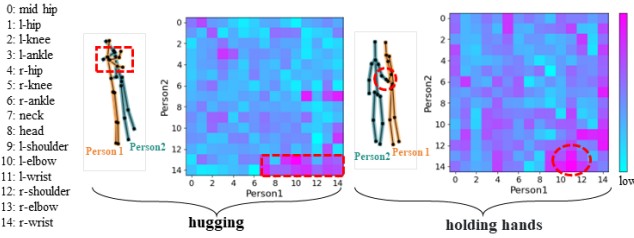

**Figure 6: Correlation visualization between two interacting individuals. The large correlation values pinpoint the key points, where their interaction is most intense.**

**Effectiveness of Hybrid Supervision**. The purpose of Hybrid Supervision Mechanism is to reduce the accumulation and propagation of errors in each rolling prediction. We employ the Hybrid Supervision Mechanism by introducing ground truth as the predicted frames. When we remove Hybrid Supervision, the performance drops substantially as in the second row of Table 5. Apparently, our model equipped with Hybrid Supervision can enhance the model's robustness to noise. Additionally, to verify the importance of the hybrid supervision in our framework, we compare the training loss and test loss on the CMU-Mocap dataset under three different training conditions with: hybrid supervision, i.e., $\mathcal{L}_{hybrid}$, standard supervision only, i.e., $\mathcal{L}_{error}$ and auxiliary supervision only, i.e., $\mathcal{L}_{gt}$. As shown in Figure 4, it can be observed that the final result with hybrid supervision (0.6) is much smaller than the result with standard supervision (0.7) and auxiliary supervision (1.7) only, indicating the effectiveness of the hybrid supervision.

**Effectiveness of SJE module**. SJE is designed to learn the spatial relationships between all joints. If the SJE is removed, the performance drops substantially as shown in the third row of Table 5. **Effectiveness of TJE module**.TJE is designed to learn the independent variations of each joint over time and captures the motion characteristics of each joint. As shown in the last row of Table 5, if the TJE is removed, the experimental results become worse.

## 6.5 Fine-grained Correlation Visualization

We visualize the attention score between individuals' query and key features in the attention mechanism in Figure 6. The figure shows the motion of two people, in the left figure, we can see person1 and person2 hugging each other, with higher attention weights on their hand and neck joints. In the right figure, as person1 takes person2's hand, the hands of person1 and person2 have higher attention weights.

## 7 Conclusion

In this paper, we present a transformer-based framework designed to effectively reduce uncertainty for multi-person pose prediction. We first introduce Hybrid-Supervised coder (HScoder) block that provides rolling guidance for multi-frame prediction. We also develop a hybrid supervision mechanism to ensure the training efficiency and the robustness of the final learned model to the accumulated prediction errors. In addition, we integrate the STE block into HScoder for fine-grained modeling to learn spatiotemporal correlations. Experiments demonstrate that our method outperforms state-of-the-art methods on multiple motion datasets.

## Acknowledgments

This work was supported by National Natural Science Fund of China (62176064) and Shanghai Municipal Science and Technology Commission (22dz1204900).

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
