# OpenReview forum: "Future Motion Dynamic Modeling via Hybrid Supervision for Multi-Person Motion Prediction Uncertainty Reduction"
_acmmm.org/ACMMM/2024/Conference — MM2024 Poster_

### Official Review · Reviewer_cj2m · 2024-04-30

**Rating:** 4
**Confidence:** 3

**Summary:**

The manuscript "Future Motion Dynamic Modeling via Hybrid Supervision for Multi-Person Motion Prediction Uncertainty Reduction" presents a novel approach to the problem of multi-person motion prediction. The authors introduce the Hybrid Supervision Transformer (HSFormer), a method that addresses the challenge of uncertainty in predicting future movements of individuals in a group. The proposed model incorporates a rolling prediction process and a hybrid supervision mechanism to enhance the robustness of predictions and mitigate the accumulation of errors. The manuscript also introduces a Spatio-Temporal Encoder (STE) block to capture fine-grained spatio-temporal correlations, which is crucial for improving prediction accuracy. The method has been evaluated on multiple datasets and has demonstrated state-of-the-art results in both short- and long-term predictions.

**Strengths:**

[-] The introduction of the Hybrid Supervision Transformer (HSFormer) is a significant contribution to the field of human motion prediction.

[-] The concept of hybrid supervision is particularly novel and addresses a key challenge in the domain.

[-] The paper is well-structured, with clear explanations of the methodology and the mathematical formulations underlying the proposed model.

[-] The paper demonstrates that the model is robust to input deviation, which is a critical aspect when dealing with real-world applications where data can be noisy or incomplete.

**Limitations:**

[-] In the introduction, the author should summarize the limitations of the existing work.

[-] the contributions of the proposed approach should be strengthened by highlighting the unique contributions of each component rather than merely describing them. Each component should be presented in terms of its specific contribution to the overall approach.

[-] While the model performs well on the evaluated datasets, it is unclear how it would generalize to scenarios with different types of motion or in the presence of occlusions.

[-] Addressing the specific challenges and potential improvements for long-term prediction would make the manuscript more comprehensive.

[-] More diverse datasets should be considered.

**Suitability:**

3

---

### Official Review · Reviewer_9kzh · 2024-05-21

**Rating:** 4
**Confidence:** 4

**Summary:**

This paper focuses on the multi-person human motion prediction. It designs a novel spatial-temporal Transformer module that explicitly  models the motion dynamics by constructing the interdependence among the adjacent frames. Furthermore, it proposes a hybrid supervision mechanism that effectively reduce the overfitting problem.

**Strengths:**

- The proposed model is novel and effective, the emphasis on motion dynamic modeling is well motivated and is proven to be effective.

- The concept of hybrid supervision is both intriguing and technically sound. It has the potential to introduce fresh perspectives to the community

- The experiments are comprehensive and solid.

**Limitations:**

- The writing should be further improved, the formula definition and mathematical notations are quite messy. For example, in Eq (1), how the neighbor motion dynamics $N^{l_d} \in \mathbb{R} ^{N \times l \times J \times C}$ construct the query $Q^{l_d} \in \mathbb{R} ^{N \times J \times C}$ with a 1-D conv? Applying a  Conv1d will change the length of tensor dimensions, and also the last dimension should be the length, not the channel. In addition, the definition of neighbor frames $N^{l_d}$ in Eq (3) and history context $H^{l_d}$ in Eq (4) should be introduced before Eq (1). Same problem appears in Section 5.2, some equations are not even with an index number

- I'm wondering how the "hybrid supervision mechanism" compares with standard data augmentation technics in motion prediction, such as motion sequence reverse https://arxiv.org/abs/2110.07495

- Since this paper focuses on multi-person motion prediction, I'm curious how this model performs on multi-person dataset with high interaction, such as ExPI dataset in ref [15].

- Typos. $X_r^n$ -> $X_{i, r}^n$ in APE definition.

- This work present a unimodal application, i.e. predicting the 3d human motion skeleton based on historical observation, which may not be very consistent with the theme of ACM MM

**Suitability:**

2

---

### Official Review · Reviewer_UhSY · 2024-05-24

**Rating:** 3
**Confidence:** 4

**Summary:**

The authors proposed a Hybrid Supervision Transformer with hybrid supervision mechanism to address the problem of multi-person motion prediction.

**Strengths:**

### Evaluation:

The quantitative results presented in the paper look pretty good.

### Technical Correctness:

The technical details sounds correct in the main paper.

**Limitations:**

### Novelty

(1) The STE block looks to be a product of the stitching of TBIFormer[1] (TBPM), motion-transformer[2] (spatial-temporal attention), which contributes a little and be over-claimed in the Intro section.

(2) I am very confused about the motivation behind the Auxiliary Supervision. One of the biggest challenges of the prediction problem is resolving uncertainty. Leaking ground-truth directly in early stage feels like it's all about a trick to fit datasets. Please explain your motivation for doing this. Did you test this stragety in inference stage for the unseen interaction or situation (may adopt cross-dataset validation).

### Experiments:

Although quantitative results look competitive, there are no more qualitative results on the other datasets (except for MI-Motion), even in the appendix.

### Writing and Drawing:
- Abstract and some sections need further rewording.
- The Figure 2 looks distracting and contains too much elements. I recommend authors to redraw it and improve its resolution.


### Conclusion:

Overall, some of the ideas are poorly motivated and have limited contributions. Therefore, I am inclined to reject this paper.

*Reference:*
>[1] Trajectory-Aware Body Interaction Transformer for Multi-Person Pose Forecasting;
>[2] A Spatio-temporal Transformer for 3D Human Motion Prediction;

**Suitability:**

2

---

### Meta-Review · Area_Chair_pGff · 2024-07-10

**Recommendation:** Accept (Poster)
**Confidence:** 2

**Metareview:**

The paper addresses multi person motion modelling. Reviews are mixed with one negative score, lowered after the rebuttal, one positive score raised and one kept. The use of teacher forcing does not seem to invalidate the approach and regarding novelty two reviewers are recognizing the value of the proposed approach.